# Smart home adoption factors: A systematic literature review and research agenda

**Alejandro Valencia-Arias** [1]*, **Sebastian Cardona-Acevedo**[2], **Sergio Gómez-Molina**[3], **Juan David Gonzalez-Ruiz**[4], **Jackeline Valencia**[5]

1 School of Industrial Engineering, Universidad Señor de Sipán, Chiclayo, Perú, 2 Institución Universitaria Escolme, Centro de investigaciones, Medellín, Colombia, 3 Coordinación de Investigaciones e Innovación, Fundación Universitaria Católica del Norte, Medellin, Colombia, 4 Facultad de Ciencias Humanas y Economicas, Departamento de Economía, Universidad Nacional de Colombia—Sede Medellín, Medellín, Colombia, 5 Instituto de Investigación y Estudios de la Mujer, Universidad Ricardo Palma, Lima, Peru

* valenciajho@uss.edu.pe

## Abstract

Smart homes represent the complement of various automation technologies that together make up a network of devices facilitating the daily tasks of residents. These technologies are being studied for their application from different sectors, including the projection of their use to improve energy consumption planning and health care management. However, technology adoption depends on social awareness within the scope of cognitive advantages and innovations compared to perceived risk because although there are multiple benefits, potential users express fears related to the loss of autonomy and security. This study carries out a systematic literature review based on PRISMA in order to analyze research trends and literary evolution in the technological adoption of smart homes, considering the main theories and variables applied by the community. In proposing a research agenda in accordance with the identified gaps and the growing and emerging themes of the object of study, it is worth highlighting the growing interest in the subject, both for the present and its development in the future. Until now, adoption factors have been attributed more to the technological acceptance model and the diffusion of innovation theory, adopting components of the Theory of Planned Behavior; therefore, in several cases, the attributes of different theories are merged to adapt to the needs of each researcher, promoting the creation of empirical and extended models.

## Introduction

Due to the rapid development of technologies in the fourth industrial revolution, characterized by the convergence of digital, physical and biological technologies that are transforming the way we live and work, and their application in different sectors of the economy, the Internet of Things (IoT) has become an ally for the development of infrastructures worldwide. In this way, the construction sector has benefited, through the use of the IoT, from the creation of smart home environments [1].

From the offer of houses equipped with technologies designed to provide adequate services, smart homes seek to meet the needs of each inhabitant to improve their quality of life [2]. By

**Data Availability Statement:** The data availability statement for this study has been duly recorded and archived in the Zenodo open data repository, a repository recognized for its commitment to the

accessibility and preservation of scientific data. The data and materials supported by this study are publicly available and can be accessed at the following DOI link: https://doi.org/10.5281/zenodo.8381268. This repository ensures the availability of and access to the underlying data used in research, promoting transparency and reproducibility of results, thereby enhancing the reliability and utility of scientific research.

**Funding:** The authors received no specific funding for this work.

**Competing interests:** The authors have declared that no competing interests exist.

making use of artificial intelligence, behavioral data can be used, and information on user preferences can be generated [3], enabling inhabitants to monitor and control a wide range of household appliances remotely and intelligently [4]. That is, smart homes represent a set of technologies that provide a human-centered network environment to connect hardware and applications in the home [5] that facilitate communication and collaboration between different devices by offering five main types of services: support, monitoring, provision of therapy, provision of comfort and counseling [3].

However, although the concept has gained popularity recently, it is not a new term because it has been discussed since the 1980s, evolving from traditional home automation to the present, where large global companies such as Google, Amazon, and Samsung Electronics are offering innovative products and services to take advantage of market growth [6]. In addition, it is estimated that cities will face an urban transformation in the coming years to manage their resources; therefore, the concept of smart cities involves various challenges related to sustainability, transport, the economy, governments, and lifestyles [7].

This industry is considered one of the most promising thanks to the rapid development of mobile network infrastructure [5]. In addition, smart homes are conceived as an option for energy management planning and health care management, and their technological adoption depends on the public's perception of the benefits and perceived innovation [8]. This should be considered in conjunction with perceived risks because although there is a positive perception of smart home implementation, there is also a certain fear linked to the loss of autonomy and privacy at home [9,10], which limits perceived value and requires companies to work on the development of proposals related to safety [11].

According to the above, despite the potential of these systems to improve quality of life and experience, users' acceptance of smart homes still does not meet expectations [4]. In the case of health care management, although its application reduces the cost of healthcare, smart technology has not spread because adoption is very low. In part, due to an inadequate understanding of the expectations, needs, and preferences of the users, and taking into account that the main audience is elderly individuals; this represents a significant challenge for successful implementation due to the conservative vision and technological concerns of this type of user [12,13]. Likewise, the popularity of smart homes is growing slower than expected and must be studied from the demand perspective [14].

Related studies show that expected performance, social impact, and cost are significant predictors of smart home adoption [15] and that perceived security risk affects the intent to use [16]. Therefore, companies dedicated to offering these services have the challenge of generating mechanisms that allow users to generate trust from controlling their domestic data, not only to overcome current restrictions but also to help people maintain a commitment to home life [17]. Considering that decision-making is not linear but unfolds through different stages, persuasion can begin as soon as consumers become accustomed to the technology, and knowledge can be developed after they have shown interest and decided to use it [18].

In accordance with the above, it is necessary to carry out a systematic review due to the growing interest in scientific production related to the subject, where various models have emerged to validate the level of technological adoption of smart technology homes [19]. Furthermore, considering the different perceived uses and the possible target audiences, there are also studies on programming optimization models to reduce the peak load and the cost of electricity [20], among other tools that seek to increase the use value of these technologies. However, to the best of our knowledge, no study in the current literature has provided an in-depth analysis of smart homes and services. To help bridge the identified knowledge gap, this study aims to conduct a systematic literature review in order to have a better understanding on this topic.

It allows to consider its relevance at present as well as the level of effect on society and its behavior [21] and seeking to identify the present context in which they operate, including the benefits and challenges generated from their technological adoption in the market, the perception of users and possible future scenarios. This systematic literature review focuses on the following research questions:

**PI1:** What are the leading research trends that address the adoption of smart homes?

**PI2:** What is the evolution of the main keywords in the knowledge body on smart home adoption?

**PI3:** What are the main theories used by researchers to determine the adoption of smart homes?

**PI4:** According to the knowledge body, what are the main variables used to understand the adoption of smart homes?

**PI5:** What research gaps are identified, and what further research questions can be formulated from these?

**PI6:** What elements should a research agenda have that integrate the identified gaps and the growing and emerging research themes on the adoption of smart homes?

Based on these research questions, the other sections of this study are composed as follows. First, the following section explains the materials and methods employed to answer the questions. Subsequently, using the studies identified to respond to the first four questions, the results section analyzes trends in bibliometric terms such as keywords, theories, and main variables in the adoption of smart homes. Then, the results are discussed, in which comparisons are made with other studies. Next, the main limitations and research gaps identified are detailed based on the previously described results to formulate a series of questions for further research as well as a research agenda, thus providing answers to questions five and six. Finally, after indicating the guidelines for further research, a discussion is presented with the achievements obtained about the proposed objective and the main conclusions of the systematic literature review on the adoption of smart homes.

## Materials and methods

To meet the objective of this research and answer the questions posed in the introductory section, this systematic literature review is conducted using an exploratory and descriptive methodology through which it is possible to understand the state-of-the-art of field of research, through which it is possible to recognize, on the one hand, limitations in existing studies and, on the other hand, guidelines for further research [22,23].

For this systematic literature review, items described in the international PRISMA statement (Preferred Reporting Items for Systematic Reviews and Meta-Analyses), 2020 version, as seen in [24], and the methodological guidelines described in [25] were followed with regard to reporting the materials and methods, inclusion criteria, exclusion criteria, information sources, search strategies, data management, and flow chart designed by PRISMA to account for the entire methodological design.

### Inclusion criteria

Inclusion or eligibility criteria refer to all the elements to be analyzed in the systematic review process, it is initially based on the strategic relationship of a series of keywords. Specifically, for this research on the adoption of smart homes, all the studies that, both in the titles and in the

keywords, contain the concepts of Smart home, Smart house, Intelligent home, or Intelligent house or Home automation, validated by specialized engineering thesauri such as the IEEE, are included. Likewise, to fully address the object of study, these studies include the concepts of adoption or acceptance in the same metadata as title and keywords.

The inclusion criteria used in this systematic review are based on previous studies on smart home adoption, such as [26,27], which conducted systematic literature reviews on smart home adoption in different dimensions. Both studies used a set of keywords such as "smart home", "smart house", "smart home", "intelligent house", "home automation", and "adoption" in the titles and keywords, which were validated by specialized engineering references such as the IEEE.

## Exclusion criteria

Exclusion criteria refer to the aspects that are considered to exclude studies during the systematic review process. According to the PRISMA 2020 statement, the exclusion criteria are applied in three consecutive exclusion phases. In the first phase, all studies that do not address the pre-defined research questions according to the title and abstract are excluded. In the second phase, all studies that do not provide access to the full text are excluded, making it impossible to analyze the models and drivers of smart home adoption in this case. These first two phases of exclusion were established from a previous bibliographic review, which found similarity of criteria in research such as that of [26,28].

Finally, once the resulting full texts have been analyzed, all scientific records that do not identify smart home adoption models, which is a fundamental purpose of the research, are excluded, leaving only scientific studies that provide answers to the questions raised in this study, deriving this exclusion criterion from the scientific literature, in articles such as those of [26,27].

## Information sources

According to the PRISMA statement for literature reviews, once the inclusion and exclusion criteria are defined, it is necessary to specify the sources of information from which the inputs that will be subjected to detailed analysis will be extracted. In this sense, for this systematic literature review, Scopus and Web of Science are selected as sources of information because, as evidenced in [29], they are currently the two most important sources of bibliographic information in terms of supplying scientific metadata and carrying out bibliometric indicators that allow for a consistent evaluation of the scientific activity reflected in the literature.

## Search strategy

After defining Scopus and Web of Science as the sources of information, it is essential to design a consistent search strategy that accounts for the previously detailed eligibility criteria so that the studies retrieved from both databases are directly related to the purpose of the research and for the characteristics of the search interface of each database. Therefore, the following specialized search equations were designed:

Scopus: TITLE ("smart house" OR "smart home" OR "intelligent home" OR "intelligent house" OR "home automation") AND TITLE (adoption OR acceptance)) OR (KEY ("smart house" OR "smart home" OR "intelligent home" OR "intelligent house" OR "home automation") AND KEY (adoption OR acceptance)).

Web of Science: (TI = ("smart house" OR "smart home" OR "intelligent home" OR "intelligent house" OR "home automation") AND TI = (adoption OR acceptance)) OR (AK = ("smart house" OR "smart home" OR "intelligent home" OR "intelligent house" OR "home automation") AND AK = (adoption OR acceptance)).

These search equations were designed to identify studies related to the adoption and acceptance of technologies related to smart or automated homes. Both equations search for key terms related to smart homes, such as "smart house", "smart home", "intelligent home" and "home automation", and combine them with terms related to the adoption and acceptance of these technologies, such as "adoption" and "acceptance". The goal is to retrieve research that examines factors that influence the adoption of smart home technologies, as well as users' perceptions and attitudes toward these technologies.

## Risk of bias assessment

The assessment of risk of bias in the studies included in this review followed a rigorous and consistent process. To ensure the quality and integrity of the results, all authors participated in the data collection and risk of bias assessment. An automated tool based on Microsoft Excel® was used to ensure an objective and standardized assessment. Each study was thoroughly reviewed by the authors independently, and any discrepancies or concerns were addressed collaboratively until consensus was reached. This risk of bias assessment methodology ensures the reliability of the results by applying consistent and transparent criteria throughout the scientific literature review process.

It is important to recognize that there is a bias in the sources of information used in this research, as they were limited to Scopus and Web of Science as databases to search for studies. This choice may have omitted relevant studies that may be available in other information sources or in more specialized databases related to smart homes. Despite this limitation, steps were taken to mitigate this bias by using the previously described inclusion and exclusion criteria and by conducting a comprehensive search within the databases selected from the search equation. However, it is important to note that there may still be relevant studies in other sources not included in this study, which could affect the breadth of the review and the generalizability of the results.

## Data management

The application of the search strategies in each database allowed the initial retrieval of 239 scientific studies related to the adoption of smart homes, of which 189 were obtained from Scopus database and 50 were obtained from Web of Science database. These studies were exported to and stored in Microsoft Excel®, through which a data homogenization process was carried out to unify the format because of the typological differences in providing information from both databases. The same tool was used to apply the exclusion criteria and to conduct the data analysis to answer the posed research questions.

## Study selection process

According to the PRISMA 2020 statement [24], it is necessary to define the process of selecting the studies for a systematic literature review. Each investigation author independently executed the search strategy and exclusion processes to reduce informational bias, using Microsoft Excel®. All the differences found were analyzed in strategic sections justified for each case until the information converged.

## Effect measures

In the context of a systematic literature review on smart home adoption, emphasis is placed on specifying the effect measures used in the synthesis or presentation of results. Although these measures are more common in primary research, this study is based on secondary sources and focuses on the analysis of relevant geographic contexts, target populations, psychometric theories, and key explanatory factors or variables in these theories. This analysis is facilitated by the use of tools such as Microsoft Excel®, ChatGPT®, and Google Bard®, but it is emphasized that individual verification by the authors is essential to ensure the quality and accuracy of the data. This approach provides a comprehensive and contextualized view of smart home adoption from a perspective based on systematic literature review, although specific effect measures such as hazard ratios or mean differences are not used.

## Synthesis methods

On the other hand, specific procedures were used to determine the eligibility of the studies included in the syntheses. For this purpose, three exclusion phases were carried out based on predefined criteria. Once these phases were completed, all the data collected were tabulated using the automated tool Microsoft Excel®. The data were obtained from the responses generated by the artificial intelligence tools ChatGPT® and Google Bard® and were subsequently subjected to manual validation by the authors to ensure accuracy and consistency in the preparation of the data for analysis. presentation or synthesis, and to contribute to the transparency and integrity of the systematic review process.

## Certainty assessment

In addition, an individual certainty assessment method similar to that used for primary studies is applied. This involves a detailed, full-text examination of each article included in the systematic literature review. This process allows for the identification of the information under review. However, biases in the study are reported and limitations are discussed in the discussion section of the article. In this way, a comprehensive assessment of the reliability of the collected evidence is provided, contributing to a sound understanding of the findings and their implications in the context of smart home adoption.

## Methodological design

In Fig 1, the flow chart recommended by the international PRISMA 2020 statement is presented to account for all the aspects related to the inclusion, exclusion, and definitive selection of studies to be analyzed in the present systematic literature review.

In the initial search phase, a total of 239 relevant documents were retrieved. However, during the selection and review process, 43 duplicate documents were identified and eliminated, reducing the number to 196. Exclusion criteria were then applied to refine the selection. Of these, 16 documents were not related to the topic of smart home adoption according to the established criteria. In addition, 139 of the remaining documents were found to be inaccessible in full text. This lack of access may be due to several reasons, such as publisher access restrictions, institutional subscriptions required to access certain articles, or limited online availability of certain documents. Finally, after reviewing the remaining 41 full-text articles, 29 of them were excluded because they did not present theoretical models of smart home adoption, resulting in the final selection of the 12 studies included in the systematic review.

An overview of the 12 articles included in this review is presented in Table 1. This table provides basic information about each article, such as title, authors, year of publication, target

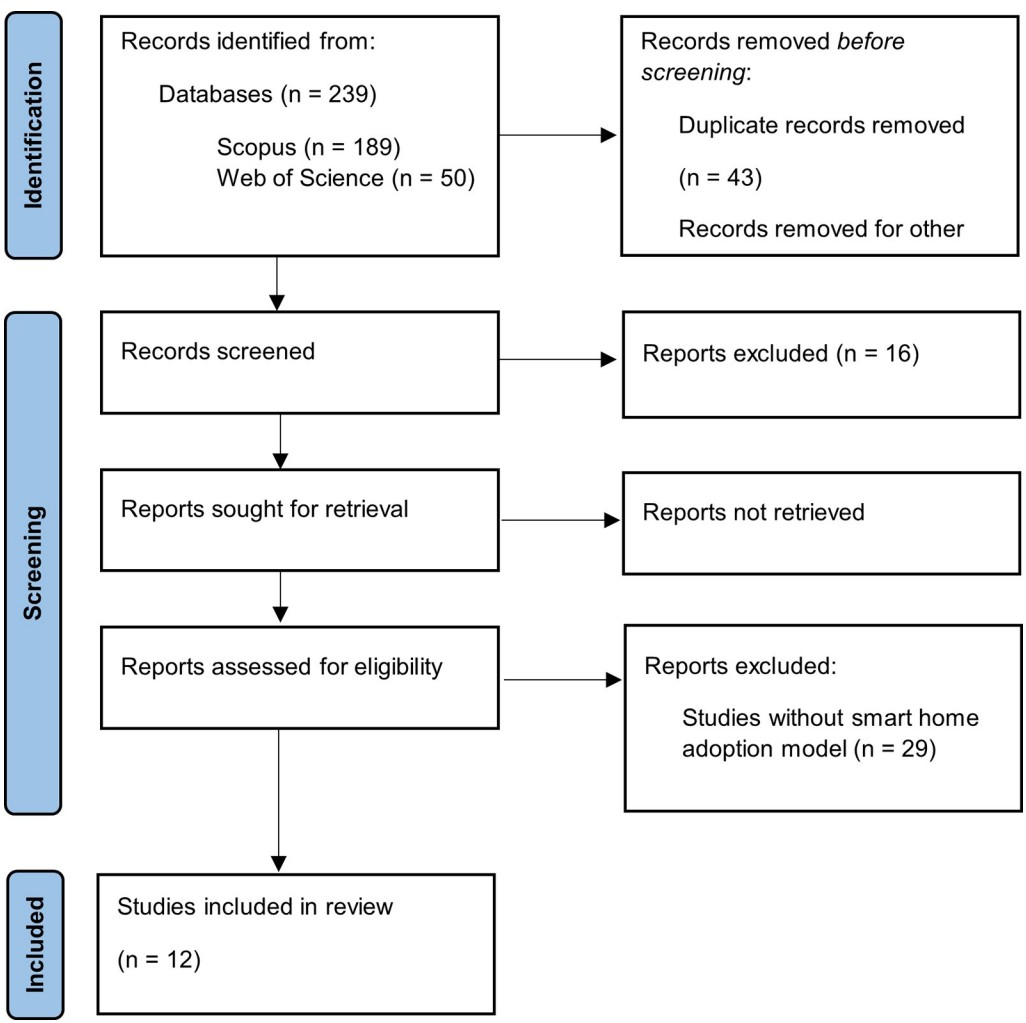

**Fig 1. Flowchart for PRISMA systematic literature reviews.**

population, country of study, and sample size. In addition, the theoretical model of smart home adoption used in each study is identified, which provides a useful reference for understanding the different perspectives addressed in the scientific literature on this topic.

## Results

This results section provides a comprehensive analysis of the findings from the systematic review of the 12 articles included in this study. Specifically, key aspects will be addressed, such as the target populations involved in the adoption of smart homes, the underlying theories that underpin this process, and the primary variables that influence this adoption. In addition, the research gaps identified in this analysis are identified and discussed in order to provide a comprehensive view of the topic and highlight areas of interest for future research.

### Populations under study

As shown in Table 1, according to the sample of articles included in this systematic literature review, smart home adoption has only been validated in three types of populations: older adults, potential consumers, and general consumers, with the latter being the more common.

**Table 1. Summary of studies included in the review.**

| N° | Article | Authors | Year | Target population | Country | Sample size | Theoretical model |
|---|---|---|---|---|---|---|---|
| 1 | Cognitive Dissonance in Technology Adoption: A Study of Smart Home Users | [3] | 2020 | General users | not specified | 387 | Own model |
| 2 | IoT Smart Home Adoption: The Importance of Proper Level Automation | [6] | 2018 | General users | South Korea | 216 | Own model |
| 3 | The influence of acceptance and adoption drivers on smart home usage | [30] | 2019 | General users | not specified | 409 | TAM; DOI |
| 4 | Senior citizens' acceptance of connected health technologies in their homes | [31] | 2019 | Elderly | not specified | 200 | TAM |
| 5 | A comprehensive acceptance model for smart home services | [32] | 2021 | General users | Jordan | 750 | TAM; DOI; TPB |
| 6 | Analyzing the Elderly Users' Adoption of Smart-Home Services | [33] | 2018 | Elderly | India; Thailand; Indonesia; Malaysia | 239 | Extended TAM |
| 7 | Factors Influencing Intention of Greek Consumers to Use Smart Home Technology | [34] | 2022 | General users | Greece | 108 | Extended TAM |
| 8 | The Investigation of Adoption of Voice-User Interface (VUI) in Smart Home Systems among Chinese Older Adults | [35] | 2022 | Elderly | China | 420 | Extended TAM |
| 9 | Privacy concerns in the smart home context | [36] | 2020 | General users | Germany | 187 | Extended TPB |
| 10 | What influences the perceived trust of a voice-enabled smart home system: An empirical study | [37] | 2021 | General users | China | 475 | Own model |
| 11 | Smart home adoption: The impact of user characteristics and differences in perception of benefits | [38] | 2021 | Potential users | South Korea | 400 | Own model |
| 12 | The heat is off! The role of technology attributes and individual attitudes in the diffusion of Smart thermostats–findings from a multi-country survey | [39] | 2021 | General users | France; Germany; Italy; Poland; Romania; Spain; Sweden; United Kingdom | 5517 | Own model |

When it comes to the adoption of smart home technologies among general consumers, several studies have shed light on the determinants of their adoption intentions. Studies such as [34] in the Greek context have identified a number of key influences on this intention. [3] have explored cognitive dissonance in the adoption of these technologies, highlighting the critical importance of user perceptions in this process. Furthermore, research by [6] has emphasized the importance of an appropriate level of automation in smart home adoption, highlighting the need for the technology to be easily manageable by consumers.

On the other hand, studies such as [30] have made significant contributions to the field by exploring the acceptance and adoption drivers that influence smart home use. [32] have presented a comprehensive acceptance model for smart home services, providing a deeper understanding of the key determinants of adoption in this context.

Furthermore, [36] have addressed privacy concerns in the context of smart homes, a fundamental aspect in the adoption decision, while [37] have explored perceived trust in voice-enabled smart home systems, highlighting its influence on adoption. Finally, [39] have investigated the diffusion of smart thermostats, highlighting the importance of technological attributes and individual attitudes in adoption. Taken together, these studies have enriched our understanding of the adoption of smart home technologies among general consumers.

With regard to the older adult population, [31] conducted research on the adoption of connected health technologies in the homes of this demographic, highlighting the importance of adapting technological solutions to specific needs. [33] analyzed the adoption of smart services in elderly homes, highlighting the importance of usability and ease of use. In addition, [35] focused on the adoption of voice interfaces in smart home systems among Chinese elderly, providing valuable insights into the preferences of this demographic. As potential consumers,

this demographic group shows noticeable differences from general consumers in terms of smart home adoption. [38] examined the impact of user characteristics and differences in perceived benefits on smart home adoption. This study highlighted the importance of understanding individual differences and benefit perceptions to develop effective marketing and promotion strategies.

## Theory analysis

The selected theories allow understanding of the adoption of smart homes through various models, as seen in Table 2, where information is provided for publications between 2018 and 2022, in particular, the models used and the main territories where they were developed. The studies were conducted mainly in European and Asian countries, which, as previously expressed, are the countries that have generated the most publications on the subject.

Six main theories are presented, for which the first is the technological acceptance model (TAM), which is addressed in [40], where they explain that the model arises as an adaptation of the theory of reasoned action (TRA) to predict the interest of users to adopt a new technology, explaining that ease of use is reflected in perceived usefulness. In addition, the TAM allows the study of the impacts of external factors that interfere in the behaviors of users, adopting the concepts of perceived usefulness (PU), perceived ease of use (PEOU), and attitudes toward the use of services (ATT), with the latter determined by the first two [31,32].

Additionally, studies were found in which the authors propose integral models of acceptance of technology as an extension of the traditional TAM, justifying that it is necessary to adapt the model based on the various particularities that consumers have for each technology [34]. A second theory is developed from these constructs, for which exogenous concepts supported by those already mentioned above are introduced. Variables related to the psychology, economic and hedonic value, usability, and security of smart homes are addressed, and hypotheses related to self-capacity, automation, universal connectivity, privacy, affordability, enjoyment, satisfaction, compatibility, and subjective norms are measured; the latter is related to the perception acquired by other people's opinions [33]. Another study incorporates the trust variable, noting the uncertainty perceived by users in relation to the computing environment and emphasizing three main dimensions: competence, related to the capabilities of a system; benevolence, referring to the belief that power will be abused; and integrity, such as appropriateness of behavior [35].

However, TAMs analyze elements related to demographic and psychographic factors of users, while the diffusion of innovation (DOI) theory focuses on studying only those factors related to technology. Although these theories make significant contributions, they have recently begun to merge into more extensive models that contribute to the general understanding of adopting smart home technologies. In this way, DOI incorporates elements that are not typical of TAMs, such as experimentation with technology and the role of technology

**Table 2. Theories of smart home adoption.**

| Theory | Country | Frequence | Authors |
|---|---|---|---|
| TAM | Germany; Jordan | 3 | [30–32] |
| Extended TAM | India; Thailand; Indonesia; Malaysia; Greece; China | 3 | [33–35] |
| DOI | Germany, Jordan | 2 | [30,32] |
| TPB | Jordan | 1 | [32] |
| Extended TPB | Germany, | 1 | [36] |
| Empirical models | South Korea; United Kingdom; China; France; Germany; Italy; Poland; Romania; Spain; Sweden | 5 | [3,6,37–39] |

in users' lives [30]. Addressing the process of innovation diffusion, from development to use, user behaviors, and user decisions, five relevant characteristics of potential users are noted: relative advantage, complexity, compatibility, possibility of proof, and observability of the innovation [32].

The theory of planned behavior (TPB) proposes that behaviors are influenced by three main beliefs: attitude, subjective norms, and behavioral control. In this sense, it takes into account scenarios where it is not possible to have general user vigilance. Therefore, the element of perceived behavioral control (PBC) is incorporated through the perspective of particular capacities based on previous practices and perceived complexity based on internal and external limitations; attitudes, on the other hand, reflect whether self-examination is conducive or not conducive to the effects of behavior; and finally, subjective norms represent the social influence on an individual [32,36].

Studies were found in which the TPB is applied in extended versions, combining its attributes with those of the TAM to evaluate danger and the intention of using smart home devices. It is relevant for companies to understand those elements that individuals consider before using their services; therefore, the TPB is used to obtain a broader representation of technological acceptance, considering PBC as the most influential variable in behavior and concluding, in turn, that behavioral control has a positive impact on intent to use [36].

Finally, there are models proposed empirically by the authors; these models provide a different perspective on adopting smart home technology based on new variables that previous theories have not addressed. For example, [38] propose within their hypothesis concepts regarding the preference for energy and health services and their possible impact on intent to use. Additionally, [6] propose that perceived controllability and interconnection are elements that should be considered to carry out measurements.

## Analysis of the main variables

The most relevant variables were identified and are detailed in Table 3, which includes recent studies from 2018 to 2022, the included variables, and the authors who have addressed those variables.

The results yielded 12 key concepts, which, in turn, are fully addressed within the theories previously analyzed, giving coherence to the present study. These variables are defined below based on the definitions established by the authors for implementation within each one of the models.

**Table 3. Main variables for the adoption of smart homes.**

| Variable | Authors |
| --- | --- |
| Perceived Reliability | [6,36–37,39] |
| Perceived Usefulness | [30–36] |
| Behavioral Intent | [30,32–35,38,39] |
| Perceived Ease of Use | [30–35] |
| Control of Perceived Behavior | [6,30,32,36] |
| Perceived Enjoyment | [3,33,34,37] |
| Perceived Interconnection | [6,32,33] |
| Subjective Norm | [33,36,37] |
| Perceived Cost | [32,34,39] |
| Perceived Automation | [6,33] |
| Attitude Toward Use | [32,33] |
| Perceived Compatibility | [32,34] |

For "perceived reliability" [37], state that the good quality of a system conveys a feeling of trust perceived by users and that the smart home industry is still in the early stages of development; therefore, perceived reliability is a key element to reducing uncertainty and includes elements such as trustworthiness, reliability, controllability, and competence.

Studies that employ TAMs include "perceived usefulness" (PU) as a determining factor for technological acceptance, i.e., the perceived value that a technology can enhance work performance. "Perceived ease of use" (PEOU) is related to the perception of effort associated with the complexity of learning and usability [30]. Likewise, concepts of the TPB are employed, for example, "control of perceived behavior", which is defined by Ajzen [41] and referenced in the work of [32] as "the perceived ease or difficulty in performing a behavior and its ability to reflect the experience as well as the expected impediments and obstacles".

The study "Analyzing the Elderly Users' Adoption of Smart-Home Services" by [33] is one of the most theoretically enriched investigations because several key concepts are defined in their work. For example, "perceived enjoyment" is the satisfaction related to the intention of using any product or service, and "subjective norm" corresponds to the opinions instilled in an individual by the environment due to a lack of knowledge about usability when a product is novel. Another concept, "automation", is one of the main elements in the technology applied to smart homes, and its adaptation to the domestic components of homes increases users' comfort.

Another key component is "perceived interconnection", defined as "the ability to work together reliably because there is a discrete manufacturer" [6], which, in addition, is related to "perceived compatibility", which serves as a hypothesis of the work in [34] and includes variables associated with human psychology, such as confidence and perceived enjoyment.

Finally, [32] reference Tornatzky and Klein [42] and refer to "perceived cost" as "the cost of a system, whether expensive or not, based on the financial resources of the user". Moreover, several studies contain it in the framework of technological adoption as a determinant of "behavioral intention." The latter is determined by attitude and perceived usefulness, and [30,33] reference Venkatesh [43] and define it as ´´´the degree to which a person has formulated conscious plans to carry out or not a certain future behavior".

### Research gaps

This study has identified the main research gaps in the available scientific literature (Table 4), proposing research questions so that future authors can carry out new studies that fill these identified theoretical and conceptual gaps. Such further studies will contribute to a more robust understanding of the factors that determine the adoption of smart homes from different social, cultural, economic, demographic, and geographical contexts.

## Discussion

In order to discuss the results obtained in this systematic literature review, the practical implications of both the bibliometric phase and the identification of research trends are addressed, as are the implications of the detailed analysis of theories and variables associated with the adoption of smart homes. Furthermore, the main limitations of the research, a comparison is made between the results obtained and the results reported by other similar studies, and finally, a research agenda is proposed that considers the research trends, the main theories and variables and, the identified gaps in the results section.

### Practical implications

The 12 studies that met the inclusion and exclusion criteria were analyzed, allowing the establishment of a perspective of the current environment in which the issue of the adoption of

**Table 4. Gaps and questions for future research.**

| Topic | Identified gap | Questions for further research |
|---|---|---|
| Theories of adoption and use of technology | 1. The existing research has focused predominantly on the main models of adoption, such as the TAM, DOI, and TPB, with limited attention given to one of the leading models of adoption and use of technology, i.e., the UTAUT. | QFR1. What is the role of the main factors of the UTAUT model in the adoption of smart homes? |
| | | QFR2. What are the main moderating variables of the UTAUT model in the adoption of smart homes? |
| Smart home adoption variables addressed | 1. Although perceived reliability is currently positioned as the most important variable of the adoption of smart homes, the main theories that allow understanding the variables that determine perceived reliability in the adoption of smart homes are unknown. | QFR1. What theories allow a better approach toward understanding the variables that determine perceived reliability in adopting smart homes? |
| | 2. The most frequent variable in the adoption of smart homes is perceived reliability; however, the main variables that determine perceived reliability in the adoption of smart homes in the context of emerging economies are still developing. | QFR2. What are the main variables that determine perceived reliability in adopting smart homes in contexts of emerging countries? |
| | 3. Among the variables identified in the scientific literature, attitude toward use has received little attention from researchers. | QFR3. What is the role of perceived reliability in attitude toward use by users for the adoption of smart homes? |
| Context of smart home adoption | 1. Studies on the adoption of smart homes address the context of more developed countries, omitting the variables that affect the adoption of this technology in homes in emerging countries. | QFR1. What is the difference between the main adoption variables between developed and emerging countries? |
| | | QFR2. What is the role of variables such as perceived cost and other economic and fial needs in emerging economies? |
| Other populations | 1. There is a lack of research on the adoption of smart home technologies by people with disabilities, despite the potential impact on their quality of life. | QFR1. How can smart home technologies be designed and adapted to meet the needs of people with disabilities? |
| | 2. Research tends to focus on urban environments, leaving a gap in understanding smart home adoption in rural communities where needs and challenges may be different. | QFR2. How can smart home technologies address the unique needs of rural communities and overcome infrastructure limitations? |
| | 3. Most studies have examined smart home adoption among middle- and high-income groups, leaving a gap in understanding how low-income people perceive and use these technologies. | QFR3. What are the economic and accessibility barriers to smart home adoption faced by low-income consumers, and how can they be overcome? |

smart home technology is developed, further revealing the main theories and variables related to the object of study. As a result, it was possible to identify six theories and 12 main variables covered throughout the studies analyzed.

In relation to the theories analyzed, several studies used models in combination; that is, the models were applied together to take advantage of the attributes each author considered relevant to their work. For example, although the TAM is the most used, in most cases, the authors merged the model with elements of other theories, such as the innovation diffusion theory and the theory of planned behavior. This is reflected in the fact that some studies include extended models, as mentioned by [34]. Furthermore, each technology has its own characteristics. Therefore, it is necessary to adapt models based on the needs of each application, taking into account the implications that may have on the user profile of each industry. In this way, smart homes to health sector are constructed using smart technology in order to monitor patients, especially older people, in their homes. This explains the interest in generating empirical models that fit these requirements and identifying which variables have not been considered within the traditional models to obtain a general perception of technological acceptance, considering variables such as health, which is perceived as a benefit.

Regarding variables, the most relevant based on the results obtained is perceived reliability. There has been a low response to adopting smart home technology compared to what was expected; although several benefits are attributed to implementing such technology, users perceive insecurity. As explained by [33], individuals do not want to place their trust in companies and give them access to their personal data. In fact, they explain that perceived security is even lower in older people who fear the adoption of new technologies, a population of potential users due to their health monitoring needs.

Therefore, the present research can help companies in this sector adopt measures based on the risks observed by individuals, aiming to improve the perception of potential users and their opinions about privacy and security factors surrounding applying this type of technology in homes.

On the other hand, the systematic review conducted provides valuable lessons for those responsible for creating policies and regulations in the area of smart home adoption. First, it highlights the need for updated regulation that addresses user privacy and security concerns. Consumer trust is fundamental to the success of this technology, so policies must ensure the protection of personal data and promote safety in the use of smart devices in the home. Policy-makers should be aware of the perceived uncertainty surrounding the adoption of this technology and work on policies that address these concerns.

Professionals involved in the implementation of smart home technology must recognize the importance of tailoring models and approaches to the specific needs of each application. This review highlights that a combination of theories and models can be effective in understanding technology acceptance in this diverse context. It also highlights the importance of the variable "perception of trustworthiness". Professionals need to actively address this user concern and design solutions that build trust and minimize perceived risk, especially in the older adult population, which is an important user group in the health care field.

Education professionals have a critical role to play in the effective implementation of smart home technology in distance and home-based learning. This report highlights the importance of addressing the privacy and security concerns of students and their families when using smart devices and systems in education. Educators should work closely with parents and care-givers to ensure that they are comfortable with the technology being used and understand how student data will be collected and protected.

This systematic review has important implications for developers of smart home technologies. First, it highlights the need to prioritize security and privacy in the design and development of these systems. The results show that the perception of trustworthiness is a critical factor for users, so developers must implement strong data security measures and provide transparency in the management of personal information.

In addition, this research highlights the importance of flexibility in system design. Since each smart home technology application may have specific needs and characteristics, developers must be willing to adapt their models and solutions to the needs of each market. Interdisciplinary collaboration, including the involvement of security and ethics experts, is essential to effectively address user concerns and develop solutions that inspire trust.

In terms of populations, the lack of research focused on specific populations, such as people with disabilities or rural communities, highlights the need to design inclusive and accessible technology solutions. Technology companies and product designers must carefully consider the diverse needs and limitations of these populations when developing smart home devices and systems. This includes not only adapting the user interface and functionality, but also ensuring that the solutions are affordable and easily accessible to everyone.

In addition, research gaps in low-income and minority communities pose significant challenges to technology equity. To address these gaps, it is essential that technology companies work with local organizations and communities to understand specific needs and overcome economic and cultural barriers. Not only can this lead to a larger and more diverse market, but it can also help improve the quality of life for these communities by providing access to technologies that can increase energy efficiency, safety, and comfort in the home.

## Theoretical contributions

The theoretical contributions derived from this systematic review are significant and shed light on several aspects related to technology adoption in smart homes. First, a variety of theories and models have been identified that have been applied to understand this phenomenon in different contexts and territories. In addition to traditional models such as TAM and DOI, extended and enriched models have been proposed that incorporate additional elements to address the complexity of smart home adoption.

Adapting models to the specific needs of each application is an important contribution. This review notes that there is no one-size-fits-all approach, as smart home technologies vary in their characteristics and requirements. Researchers and practitioners now have the basis to adapt theoretical models to the specifics of each industry, resulting in a more accurate and effective approach to assessing technology acceptance.

Furthermore, the importance of the variable "perception of reliability" in the adoption of smart home technology has been highlighted. This variable, which addresses the user's perception of security and reliability, was highlighted as critical in a context where the security of personal data is a primary concern. This theoretical contribution highlights the need for developers and companies in this sector to prioritize security and reliability in system design and implementation.

Another important theoretical aspect is the fusion of theories and models, which allows for a more holistic understanding of technology adoption in smart homes. Combining elements from different theories, such as integrating TAM with the Theory of Diffusion of Innovations or the Theory of Planned Behavior, provides a more complete view of the factors that influence technology adoption. This theoretical contribution promotes an interdisciplinary approach to address the complexities of technology adoption in the home.

In this sense, in terms of combining theories, Fig 2 is proposed, which presents an innovative and comprehensive theoretical model. This model combines the main theories identified in this systematic review, integrating elements of the Technology Acceptance Model (TAM), the Diffusion of Innovations Theory (DOI), and the Theory of Planned Behavior (TPB). It also incorporates key latent variables from the scientific literature. This multidimensional approach provides a solid foundation for analyzing and understanding technology adoption in the smart home from a comprehensive perspective, taking into account the complex interaction of factors that influence user decisions.

The findings of this study add valuable nuances to existing technology adoption theories and propose a new theoretical framework that reflects the complexity of technology adoption in smart homes. The proposed model, the Technology Adoption Model for Smart Homes (TAMSH), adds new nuances to traditional theories by integrating key variables specific to technology adoption in this particular context. Rather than considering only individual perceptions and attitudes toward technology, the TAMSH incorporates a broader range of influences, such as relative benefits, observability, social norms, and perceived costs. This enriches existing theories by recognizing the importance of social and contextual factors in the adoption of smart home technologies. In addition, the TAMSH highlights the importance of the variables of perceived enjoyment and perceived connectedness, which are critical to home technology adoption but often overlooked in conventional theories.

## Limitations

The studies analyzed have applied their adoption models locally; therefore, the variables associated with the findings result from the peculiarities and elements of each territory. For this reason, new empirical and extended models have been developed, which suggest that these

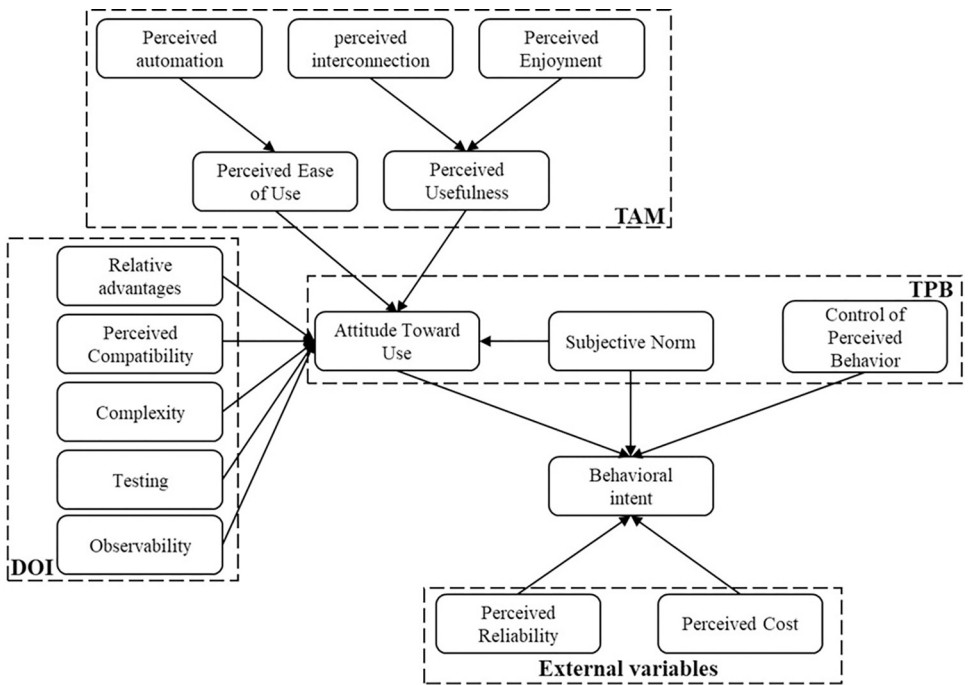

**Fig 2. Proposal for a theoretical model of smart home adoption.**

theories should be applied based on the characteristics of each sector and not in a general way, even more so taking into account the lack of studies in emerging economies because it is unknown if these same variables are applicable in that context.

It is necessary to consider that one of the main limitations of the research is incomplete retrieval, taking into account that for the study carried out, it was impossible to obtain detailed information from all the texts retrieved from the databases, and not all documents were complete or had open access to the public. Therefore, several studies were considered for the bibliometric analysis but discarded for the systematic literature review and the analysis of thematic components related to the variables and theories.

It is important to note that this systematic review has inherent limitations that must be considered when interpreting the results. One of the main limitations is the relatively small number of articles included in the final sample, which consists of 12 studies. This means that the results obtained, although significant within the selected sample, are not absolutely conclusive in terms of representing all the information available in the scientific literature on the adoption of smart homes. Future studies in this area could overcome this limitation by accessing a larger number of articles and expanding the scope of the review to provide a more complete and detailed view of the factors that influence the adoption of technologies in smart homes.

## Related work

Other studies have addressed the perception of users about the implementation of smart homes through a systematic literature review, for example, "A systematic review of the smart home literature: A user perspective", in which the authors conduct a review on the topicality of the subject from the user perspective, presenting an overview of the characteristics of smart homes, examining the impact of behavioral beliefs on user behavior and satisfaction, taking into account the perceived risks and benefits, and therefore contributing to the literature on the acceptance of technologies in private settings [44].

The approach taken in [28] involves the development of a holistic framework of smart homes from a bibliometric perspective, focusing on their intellectual structure and research trends through the application of analysis and visualization tools such as CiteSpace and VOS-viewer, identifying hot spots, current affairs and further directions of research in this field.

Additionally, [45] carried out a bibliometric and scientometric study from the perspective of users among the elderly population, analyzing research trends and identifying the need to exert greater effort to diversify funding sources and priorities. In [46], the factors that hinder and promote the adoption of smart home technologies are examined from the business point of view. Based on an analysis of previous literature and thematic map, where constructs are evaluated as drivers and obstacles to adoption, variables such as innovation, high cost, lack of compatibility, lack of ability to test it, inability to observe it, lack of a trustworthy brand, lack of favorable conditions, support services, complexity, and technological anxiety were identified.

## Research agenda

Finally, in a complementary way, in this systematic review, a research agenda is proposed so that, in addition to the identified gaps, other researchers can guide further scientific studies based on topics considered emerging and cutting-edge, as observed in Fig 3.

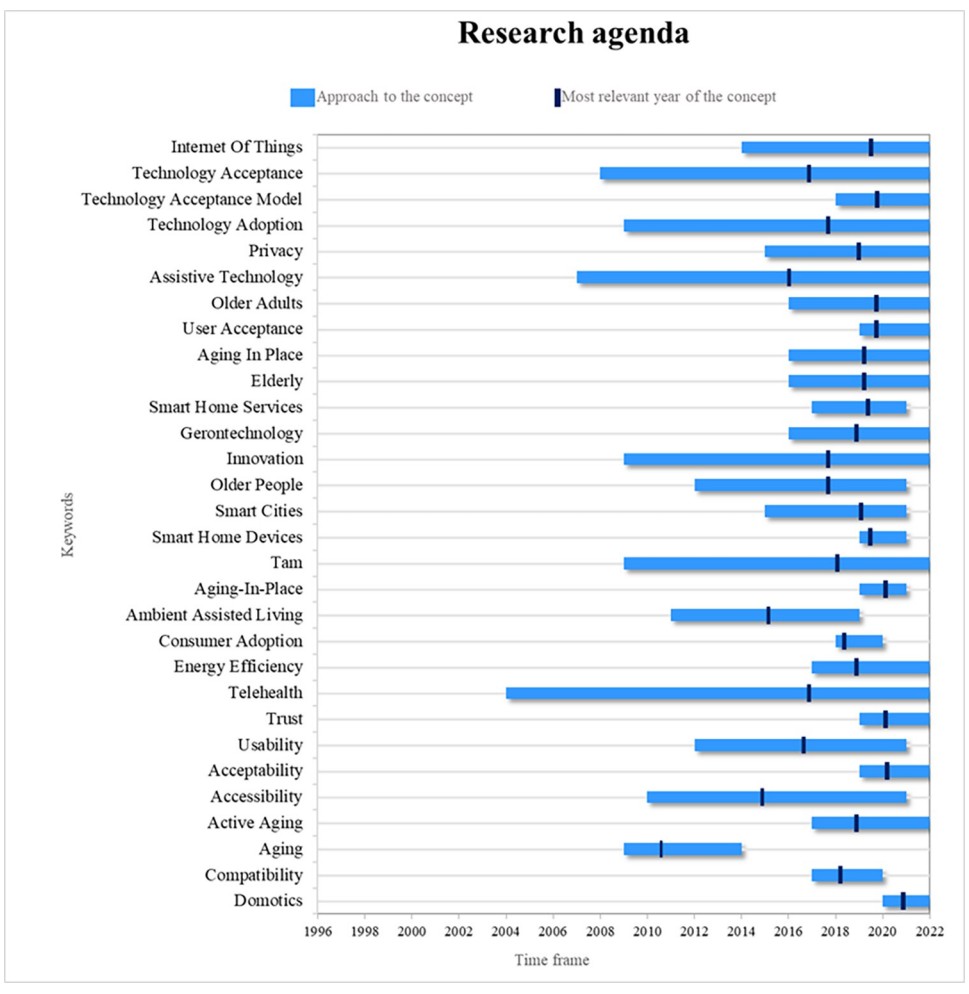

**Fig 3. Research agenda.**

The research agenda is proposed from the 30 main concepts that, secondarily, have been provided by all the authors who have researched factors for smart home adoption. The agenda displays the window of time in which the concepts have been addressed and the most relevant year, identifying the specific moment in which a concept was prominent in the scientific literature, prioritizing those concepts studied in the last year, with the most important occurring in the year closest to the present.

In this sense, it is evident that the concept of telehealth has been addressed in a greater window of time, appearing in the scientific literature approximately in 2004 and being studied to date. Therefore, although the most relevant year for this concept is approximately 2017, it is among one of the most relevant keywords, in terms of benefits, for the adoption of smart homes.

Another concept that has been transversally important for the development of the scientific body is assistive technology, which, like telehealth, is positioned among one of the most relevant subtopics in regard to understanding factors for the adoption of smart homes because it offers a greater range of possibilities for the medical care of patients. Therefore, further studies should expand on the importance of this factor, inquiring about the importance of assistive technology on perceived usefulness, which is positioned as the second most important factor in understanding the adoption of smart homes.

For understanding the factors that influence, in different populations, the adoption of smart homes, the scientific literature has made use of technological acceptance and technological adoption models in general. These two concepts are currently widely addressed. Further research can add to the understanding of the field using other models validated in the scientific literature, such as the UTAUT, as well as its multiple extensions, which, as previously identified, has been rarely addressed by different researchers, thus offering a wide margin of growth and prominence for the near further for the topic of the adoption of smart homes.

However, the UTAUT model, as well as its extensions, is not the only model that can be employed soon to understand better the factors for adopting smart homes. The model of technological acceptance (MTA) is well-positioned as the most important model to understand the factors that affect the acceptance or adoption of smart homes in different populations that, until now, have not been studied in the scientific literature.

From another perspective, the adoption of smart homes involves not only perceived usefulness, which has been addressed in telemedicine and technologies for medical care of different types of patients but also the individual innovation of users at the forefront of technology. This is currently one of the most studied concepts and further research should build on its inherent relationship with important variables such as perceived compatibility, perceived automation, and perceived interconnection, among others, to expand the knowledge of the main factors of adoption of smart homes by this type of consumer.

One of the main concepts that has emerged in the research field in the previous decade and constitutes one of the main factors today is privacy, associated with aspects of safety and reliability by consumers. Therefore, as identified among the main gaps, further research should expand the concrete understanding of the main factors that explain perceived reliability and develop solutions that allow this technological innovation to be adopted by more people to enjoy its different benefits and facilities.

Finally, from the most important technical aspect, further research should provide new perspectives for the understanding, acceptance, and use of the IoT and home automation, with the former being fundamental for the interconnection of different elements and the latter being the automation of different patterns so that different types of populations can exploit smart home technology.

## Conclusions

Based on the research findings, the adoption of smart homes is positioned as a growing theme in the scientific literature, whose bibliometric research trends show the importance of scientific and technological development in the present and the near future. Studies are growing at an exponential rate, primarily in the context of developed countries such as the United States, Germany, the United Kingdom, and South Korea, which have expanded the scientific literature thanks to both their scientific and research as technological and innovation capacity.

The thematic evolution analysis revealed that the adoption of smart homes not only relates to factors associated with health and telemedicine but also currently accounts for aspects of innovation based on consolidated technologies such as the IoT, and other emerging concepts, such as home automation, which involves aspects associated with interconnection and automation.

In addition, currently, the adoption of smart homes is more associated with technological acceptance factors than behavioral factors, with the main models being the TAM, as well as its different extensions, and the DOI, even more so than the TPB, which focuses on more psycho-behavioral aspects than on technology adoption.

It is evident that, with the TAM as the most used theory to understand the factors for the adoption of smart homes, perceived usefulness and perceived ease of use are positioned among the variables that most explain this technological adoption. However, perceived reliability is the most important variable at present and the most important shortly because security and privacy directly affect how users perceive the reliability of this cutting-edge technology, in addition to other factors such as perceived enjoyment and perceived cost.

Understanding that the main theories used to investigate the adoption of smart homes involve technological acceptance issues rather than behavioral issues, further research should add new analysis elements through one of today's leading theories, i.e., the UTAUT, as identified as a main gap in existing research.

Finally, this systematic literature review allowed for establishing the orientation for further research based on a research agenda that includes cutting-edge and emerging concepts within the research field and the main research gaps identified to expand the understanding of the factors for the adoption of smart homes from the scientific perspective and develop new technical and innovative solutions from the technological perspective.

## Supporting information

**S1 Checklist. PRISMA 2020 checklist.**
(DOCX)

## Author Contributions

**Conceptualization:** Alejandro Valencia-Arias.

**Data curation:** Alejandro Valencia-Arias, Jackeline Valencia.

**Formal analysis:** Alejandro Valencia-Arias, Sebastian Cardona-Acevedo, Juan David Gonzalez-Ruiz.

**Investigation:** Sebastian Cardona-Acevedo, Jackeline Valencia.

**Methodology:** Sebastian Cardona-Acevedo, Sergio Gómez-Molina, Juan David Gonzalez-Ruiz.

**Project administration:** Sergio Gómez-Molina.

**Resources:** Alejandro Valencia-Arias.

**Software:** Sebastian Cardona-Acevedo, Sergio Gómez-Molina, Juan David Gonzalez-Ruiz, Jackeline Valencia.

**Supervision:** Sergio Gómez-Molina.

**Visualization:** Jackeline Valencia.

**Writing – original draft:** Alejandro Valencia-Arias, Sebastian Cardona-Acevedo.

**Writing – review & editing:** Alejandro Valencia-Arias, Sergio Gómez-Molina, Juan David Gonzalez-Ruiz, Jackeline Valencia.

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
