## [Decision Letter · Decision Letter 0]

31 Jul 2023

PONE-D-23-13511Smart home adoption factors: A systematic literature review and research agendaPLOS ONE

Dear Dr. Valencia-Arias,

Thank you for submitting your manuscript to PLOS ONE. After careful consideration, we feel that it has merit but does not fully meet PLOS ONE’s publication criteria as it currently stands. Therefore, we invite you to submit a revised version of the manuscript that addresses the points raised during the review process.

We look forward to receiving your revised manuscript.

Kind regards,

Mohammed A. Al-Sharafi

Academic Editor

PLOS ONE

Journal Requirements:

Additional Editor Comments:

Dear Authors,

I have carefully reviewed your manuscript, as have our three expert reviewers. Based on the reviewers' comments and my own evaluation, your paper provides a significant contribution to the understanding of smart home adoption factors. However, there are a few areas that require attention to strengthen the manuscript before it can be considered for publication. I hope you find these comments constructive and useful for revising your paper.

1. Please consider adding a new section "Related Work" that discusses previous systematic literature reviews (SLRs) conducted on smart home adoption. This will aid in contextualizing your study's contribution to the existing body of knowledge and give readers a clear understanding of how your work adds to or builds upon prior research.

2. It's essential to provide evidence of the reliability and validity of the inclusion and exclusion criteria used in your literature search. A robust and comprehensive literature search underpins the findings of any systematic review, and showing how your criteria were developed and applied will enhance the transparency and rigor of your study.

3. The reviewers and I suggest that the manuscript should address the practical implications of your findings for policymakers, practitioners, and other stakeholders. Such insights will make your study more valuable to a broader audience, not only to academics but also those who are directly involved in the development and implementation of smart home technologies.

4. Ensure that the manuscript sufficiently addresses the theoretical contributions of your study. The application of the Technological Acceptance Model, the Diffusion of Innovation Theory, and the Theory of Planned Behaviour to the context of smart home adoption is commendable. However, further discussion is required on how your study contributes to advancing our theoretical understanding of the topic. Highlight how your findings add nuance to these theories or propose new theoretical frameworks or constructs.

As you revise your manuscript, remember that you only need to include the reviewers' suggested references if they're directly relevant to your study. Unnecessary references may dilute the focus of your work. Looking forward to your revised manuscript.

Reviewers' comments:

Reviewer's Responses to Questions

**Comments to the Author**

1. Is the manuscript technically sound, and do the data support the conclusions?

Reviewer #1: Yes

Reviewer #2: Yes

Reviewer #3: Partly

2. Has the statistical analysis been performed appropriately and rigorously? 

Reviewer #1: Yes

Reviewer #2: Yes

Reviewer #3: No

3. Have the authors made all data underlying the findings in their manuscript fully available?

Reviewer #1: Yes

Reviewer #2: Yes

Reviewer #3: No

4. Is the manuscript presented in an intelligible fashion and written in standard English?

Reviewer #1: No

Reviewer #2: Yes

Reviewer #3: Yes

5. Review Comments to the Author

Reviewer #1: Dear Author,

The researchers stated that smart homes represent the complement of various automation technologies that together make up a network of devices facilitating the daily tasks of residents. These technologies are being studied for their application from different sectors, including the projection of their use to improve energy consumption planning and health care management. However, technology adoption depends on social awareness within the scope of cognitive advantages and innovations compared to perceived risk because although there are multiple benefits, potential users express fears related to the loss of autonomy and security. This study carries out a systematic literature review based on PRISMA in order to analyze research trends and literary evolution in the technological adoption of smart homes, considering the main theories and variables applied by the community. In proposing a research agenda in accordance with the identified gaps and the growing and emerging themes of the object of study, it is worth highlighting the growing interest in the subject, both for the present and its development in the future. Until now, adoption factors have been attributed more to the technological acceptance model and the diffusion of innovation theory, adopting components of the Theory of Planned Behavior; therefore, in several cases, the attributes of different theories are merged to adapt to the needs of each researcher, promoting the creation of empirical and extended models. However, the research paper demonstrates low level of understanding of the relevant literature in the field and did not cite an appropriate range of literature sources. Methodology is weak; more rigorous analyses is needed along with the research objectives. Analyses and findings are presented in a weak manner as to present new ideas. Also, the research has not proper discussion. Though, the paper needs improvements in order to meet the standards of this journal.

Reviewer #2: This paper provides a comprehensive literature review into the relevant domain of smart home adoption factors, and stands justified by the interesting observation that the current growth of concerning technologies are not meeting the expectations of previously set forecasts, as well as that there are currently many reservations that unaware individuals have of these technologies. There is a welcomed contribution in that they articulate a problematization of the field through research gaps identified in table 3, consequently following from a derivation of key concepts in table 2.

While the paper is fundamentally warranted on these grounds, the following points need to be addressed:

On line 37, consider providing a context for readers who are unfamiliar with the advent of the 'fourth industrial revolution' - a sentence or two should be helpful.

On line 67, you relate that 'its application reduces the cost' - is this the cost of the concerning health care management strategy? Consider rephrasing this.

There is an unhelpful overlap of the same concept conveyed twice when discussing the perceived risk of smart home adoption, as follows:

* Near line 62: "there is also a certain fear linked to the loss of autonomy and privacy at home"

* Near line 75: "distrust, anxiety about technology, and potential perceived risks such as threats to privacy and security"

Consider combining the ideas of these disjoint parts into a single discussion point.

With regard to the development of the search strategy, it can be difficult for the lay reader to interpret the search equations provided, especially given that the nature of the language used to implement the search criteria is not explicitly articulated as a footnote or annotation of the strategy. For this reason, the reader would benefit from a colloquial description of the strategy eg. 'we searched documentation for which the keywords (presumably) contained the clauses "smart house", "smart home", etc.'

It is recommended that these points be addressed before the paper is accepted for publication.

Reviewer #3: Review: Smart Home Adoption Factors: A systematic literature review and research agenda

Introduction

I am pleased to provide my comments on the paper titled "Smart Home Adoption Factors: A Systematic Literature Review and Research Agenda." This study ambitiously explores the multi-faceted aspects of smart home technology adoption, providing a comprehensive overview of the topic's historical and current perspectives. The authors should be lauded for their systematic approach in using the PRISMA model to conduct their literature review, which allows for a thorough and meticulous inspection of existing research trends in this area.

The paper seeks to identify the key drivers and barriers to adopting smart home technology. This area has gained significant attention due to digital transformation and the rising demand for energy efficiency and advanced healthcare solutions. The paper's core argument centres on technological feasibility, societal perception, and the individual's perceived risks and benefits, revealing a complex interplay that ultimately shapes adoption behaviour.

The authors draw upon established theories, such as the Technological Acceptance Model, the Diffusion of Innovation Theory, and the Theory of Planned Behaviour, while recognising the need to adapt or expand to suit the unique context of smart home adoption. This provides a promising starting point but raises a few critical points that may warrant further attention.

Overall, the paper provides a foundation for understanding the determinants of smart home adoption. It opens up new avenues of research, setting the stage for further exploration in a rapidly evolving field. However, certain areas may benefit from further development and refinement to enhance the paper's utility for future research. The authors' continued work in this area is eagerly anticipated.

Major issues

Page 9, Information sources: The authors argued that SCOPUS and the Web of Science are "currently the two most important bibliographic" sources. However, this argument means the authors only looked for indexed papers within these two databases. This issue should raise a question about the exclusion criteria they adopted. Does the adoption of this search strategy related to quality issues? Comprehensiveness? Or beliefs that other sources are not as "important"? The authors should justify not expanding their search for Journals that are yet to be indexed. The IoT field is relatively new. Therefore it is expected that new journals are not indexed and may contain unreviewed manuscripts. Limiting the search for two databases a potential bias.

Page 40, Fig. 1 (PRISMA):

1- In the second stage: there are 16 excluded papers. There needs to be a mention of the reasons for excluding these studies.

2- The third stage: there are 139 non-retrieved papers. This number represents 77% of the total number of studies. What are the reasons behind not retrieving these papers? The number is large to be ignored.

On page 11, the authors mentioned reasons for excluding the papers, which should also be included in the PRISMA flowchart with the specific number of papers excluded by reason. Also, I find it hard to understand how indexed papers in SCOPUS and the Web of Science do not provide full text. I can understand this issue for a few studies but not for 77% of the studies identified.

Pages 12 and 13: The authors present a timeline (Fig. 2) for the number of publications on the subject over the years. The regression line does not provide any useful information. I am unsure how they calculated such a correlation and regression with these few points. A simple representation as a time series would be sufficient. Moreover, since the studies are not included in the review due to the lack of access to the full text, I am also unsure how they are included in the analysis. This issue applies to the other parts, publications by countries, journals, and authors.

Pages 14 and 15 (Thematic components): Again, the number of studies reviewed are 12, yet, I find the authors refer to a larger number of studies, including those dropped due to not being accessed. Suppose the authors refer only to the 12 studies analysed. This analysis cannot be considered reliable as the number of studies is too small to represent the overall thematic analysis.

In Fig. 7, the label on the y-axis is the average year. I find this labelling confusing, given the nature of the data type to have an average. It would be more appropriate to use a median year rather than an average statistically. Also, as a best practice in data visualisation, each figure title should indicate the number of studies included (sample size). I find it hard to understand if these figures are for 12 studies only or 180 studies.

 

Table 1: The number of studies presented is eleven. Does one of these studies not have a theoretical framework?

Minor issues

Figure 7 can benefit from higher resolution as it is hard to read unless the paper is zoomed in.

ICT on page 15 is mentioned without spelling the entire definition before using the acronym.

The methodology lacks many best practices in systematic reviews. The search process shows potential bias in the datasets used for this review. The authors do not present a table summarising the results (a matrix table) that includes all the studies with data extracted from each primary study. This table is a standard practice in systematic reviews that allows the reader to follow the different aspects of each study. Moreover, as in Table 1, the number of studies is less than the final number. A comprehensive table of features extracted allows better follow-up and data summarising. A comprehensive table can be used for producing summary statistics. Another major issue is the analysis of 180 studies in certain parts and then shifting to the primary 12 studies in other parts. The discussion is easy to follow and well-written.

The effort in this paper is appreciated. However, it needs improvement on many levels. Therefore, I feel it should not be considered for publication on this occasion. Major improvements are needed.

6. PLOS authors have the option to publish the peer review history of their article (what does this mean?). If published, this will include your full peer review and any attached files.

Reviewer #1: No

Reviewer #2: **Yes: **Abdul Karim Obeid

Reviewer #3: No

---

## [Author Response · Author response to Decision Letter 0]

12 Sep 2023

PLEASE LOOK AT THE ATTACHMENT SINCE THE RESPONSE LETTER HERE IS DISCONFIGURED DUE TO HAVING A TABLE

September 12th, 2023

Dear 

PLOS ONE – Editorial Team

Kind regards

In accordance with the suggestions of the reviewers in our article “Smart home adoption factors: A systematic literature review and research agenda”, the following changes were made, properly marked with red letters in the article:

Subject Revisor Comment Answer

General Journal requierements 1. Please ensure that your manuscript meets PLOS ONE's style requirements, including those for file naming. The PLOS ONE style templates can be found at

https://journals.plos.org/plosone/s/file?id=ba62/PLOSOne_formatting_sample_title_authors_affiliations.pdf The manuscript is verified to meet PLOS ONE requirements in terms of format and style.

 Journal requierements 2. In your Data Availability statement, you have not specified where the minimal data set underlying the results described in your manuscript can be found. PLOS defines a study's minimal data set as the underlying data used to reach the conclusions drawn in the manuscript and any additional data required to replicate the reported study findings in their entirety. All PLOS journals require that the minimal data set be made fully available. For more information about our data policy, please see http://journals.plos.org/plosone/s/data-availability.

We will update your Data Availability statement to reflect the information you provide in your cover letter. After the conclusions section, a "Data Availability Statement" section is added, publishing the data in the Zenodo open repository, yielding the following DOI: https://doi.org/10.5281/zenodo.8336122

 Journal requierements 3. Please include captions for your Supporting Information files at the end of your manuscript, and update any in-text citations to match accordingly. Please see our Supporting Information guidelines for more information: http://journals.plos.org/plosone/s/supporting-information. Subtitles are added according to the journal's guidelines and the number of all citations is adjusted at the end of the application of all reviewer corrections

 Editor Comments 1. Please consider adding a new section "Related Work" that discusses previous systematic literature reviews (SLRs) conducted on smart home adoption. This will aid in contextualizing your study's contribution to the existing body of knowledge and give readers a clear understanding of how your work adds to or builds upon prior research. There was a section "Other studies" that accounts for what was requested. In that sense, the name of the previous section is replaced by the one recommended by the editor

 Editor Comments 2. It's essential to provide evidence of the reliability and validity of the inclusion and exclusion criteria used in your literature search. A robust and comprehensive literature search underpins the findings of any systematic review, and showing how your criteria were developed and applied will enhance the transparency and rigor of your study. References are added that justify the inclusion and exclusion criteria of the article

 Editor Comments 3. The reviewers and I suggest that the manuscript should address the practical implications of your findings for policymakers, practitioners, and other stakeholders. Such insights will make your study more valuable to a broader audience, not only to academics but also those who are directly involved in the development and implementation of smart home technologies. An additional 5 paragraphs of practical implications are added that address the dimensions recommended by the reviewers and the editor, mentioning implications for policy makers, professionals and other interested parties such as the educational context and developers of this type of systems

 Editor Comments 4. Ensure that the manuscript sufficiently addresses the theoretical contributions of your study. The application of the Technological Acceptance Model, the Diffusion of Innovation Theory, and the Theory of Planned Behaviour to the context of smart home adoption is commendable. However, further discussion is required on how your study contributes to advancing our theoretical understanding of the topic. Highlight how your findings add nuance to these theories or propose new theoretical frameworks or constructs. A section called "theoretical contributions" is added in response to what was requested by the editor. This section also proposes a theoretical model that includes the main theories and variables of smart home adoption.

Reviewers' comments Reviewer 1 The researchers stated that smart homes represent the complement of various automation technologies that together make up a network of devices facilitating the daily tasks of residents. These technologies are being studied for their application from different sectors, including the projection of their use to improve energy consumption planning and health care management. However, technology adoption depends on social awareness within the scope of cognitive advantages and innovations compared to perceived risk because although there are multiple benefits, potential users express fears related to the loss of autonomy and security. This study carries out a systematic literature review based on PRISMA in order to analyze research trends and literary evolution in the technological adoption of smart homes, considering the main theories and variables applied by the community. In proposing a research agenda in accordance with the identified gaps and the growing and emerging themes of the object of study, it is worth highlighting the growing interest in the subject, both for the present and its development in the future. Until now, adoption factors have been attributed more to the technological acceptance model and the diffusion of innovation theory, adopting components of the Theory of Planned Behavior; therefore, in several cases, the attributes of different theories are merged to adapt to the needs of each researcher, promoting the creation of empirical and extended models. However, the research paper demonstrates low level of understanding of the relevant literature in the field and did not cite an appropriate range of literature sources. Methodology is weak; more rigorous analyses is needed along with the research objectives. Analyses and findings are presented in a weak manner as to present new ideas. Also, the research has not proper discussion. Though, the paper needs improvements in order to meet the standards of this journal. On the one hand, the methodology section of the article is strengthened, adding subsections recommended by the PRISMA declaration, to provide greater rigor. Likewise, the reason why more articles are not included in the literature review process is mentioned, justifying the final 12 articles in the inability to access full text to a significant amount of data, which is mentioned in the biases. of the study, as well as the limitations. Finally, the rigor in the presentation of findings is expanded, synthesizing information and adding "theoretical contributions" through which, on the one hand, the contributions of the article are strengthened, and, on the other hand, the structure of the section is improved. Discussion, based on comments, also, from other reviewers

 Reviewer 2 While the paper is fundamentally warranted on these grounds, the following points need to be addressed:

On line 37, consider providing a context for readers who are unfamiliar with the advent of the 'fourth industrial revolution' - a sentence or two should be helpful. A short sentence is added that explains what the Fourth Industrial Revolution is

 Reviewer 2 On line 67, you relate that 'its application reduces the cost' - is this the cost of the concerning health care management strategy? Consider rephrasing this. The phrase "of healthcare" is added to clarify what is reducing the cost in this context

 Reviewer 2 There is an unhelpful overlap of the same concept conveyed twice when discussing the perceived risk of smart home adoption, as follows:

* Near line 62: "there is also a certain fear linked to the loss of autonomy and privacy at home"

* Near line 75: "distrust, anxiety about technology, and potential perceived risks such as threats to privacy and security"

Consider combining the ideas of these disjoint parts into a single discussion point. The second paragraph in question is subtracted, as it repeats the information stipulated in the previous paragraph, in accordance with the reviewer's suggestion

 Reviewer 2 With regard to the development of the search strategy, it can be difficult for the lay reader to interpret the search equations provided, especially given that the nature of the language used to implement the search criteria is not explicitly articulated as a footnote or annotation of the strategy. For this reason, the reader would benefit from a colloquial description of the strategy eg. 'we searched documentation for which the keywords (presumably) contained the clauses "smart house", "smart home", etc.' A paragraph is added after the equations, explicitly explaining what is intended with both search equations in simple and easy to interpret terms.

 Reviewer 3 Major issues

Page 9, Information sources: The authors argued that SCOPUS and the Web of Science are "currently the two most important bibliographic" sources. However, this argument means the authors only looked for indexed papers within these two databases. This issue should raise a question about the exclusion criteria they adopted. Does the adoption of this search strategy related to quality issues? Comprehensiveness? Or beliefs that other sources are not as "important"? The authors should justify not expanding their search for Journals that are yet to be indexed. The IoT field is relatively new. Therefore it is expected that new journals are not indexed and may contain unreviewed manuscripts. Limiting the search for two databases a potential bias. A section called "Risk of bias assessment" is added to the methodology, following the PRISMA-2020 guidelines, mentioning the existing limitation in the selection of information sources.

 Reviewer 3 Page 40, Fig. 1 (PRISMA):

1- In the second stage: there are 16 excluded papers. There needs to be a mention of the reasons for excluding these studies.

2- The third stage: there are 139 non-retrieved papers. This number represents 77% of the total number of studies. What are the reasons behind not retrieving these papers? The number is large to be ignored. 1. The reason for exclusion of the 16 articles mentioned is mentioned in Figure 1, referring to the previously defined exclusion criteria.

2. In the paragraph immediately following the PRISMA flowchart, a justification is added as to why such a significant figure was excluded in terms of lack of access to full text.

 Reviewer 3 On page 11, the authors mentioned reasons for excluding the papers, which should also be included in the PRISMA flowchart with the specific number of papers excluded by reason. Also, I find it hard to understand how indexed papers in SCOPUS and the Web of Science do not provide full text. I can understand this issue for a few studies but not for 77% of the studies identified. In the paragraph immediately following the PRISMA flowchart, a justification is added as to why such a significant figure was excluded in terms of lack of access to full text. Additionally, in the limitations of the study, it is added that the results are not absolutely conclusive due to the low final number of articles included in the literature review.

 Reviewer 3 Pages 12 and 13: The authors present a timeline (Fig. 2) for the number of publications on the subject over the years. The regression line does not provide any useful information. I am unsure how they calculated such a correlation and regression with these few points. A simple representation as a time series would be sufficient. Moreover, since the studies are not included in the review due to the lack of access to the full text, I am also unsure how they are included in the analysis. This issue applies to the other parts, publications by countries, journals, and authors. In accordance with the suggestions made for the bibliometric part of the present systematic review, all figures and their respective analyzes are excluded, retaining only everything derived from the 12 articles finally included according to the designed criteria.

 Reviewer 3 Pages 14 and 15 (Thematic components): Again, the number of studies reviewed are 12, yet, I find the authors refer to a larger number of studies, including those dropped due to not being accessed. Suppose the authors refer only to the 12 studies analysed. This analysis cannot be considered reliable as the number of studies is too small to represent the overall thematic analysis. In accordance with the suggestions made for the bibliometric part of the present systematic review, all figures and their respective analyzes are excluded, retaining only everything derived from the 12 articles finally included according to the designed criteria.

 Reviewer 3 In Fig. 7, the label on the y-axis is the average year. I find this labelling confusing, given the nature of the data type to have an average. It would be more appropriate to use a median year rather than an average statistically. Also, as a best practice in data visualisation, each figure title should indicate the number of studies included (sample size). I find it hard to understand if these figures are for 12 studies only or 180 studies. In accordance with the suggestions made for the bibliometric part of the present systematic review, all figures and their respective analyzes are excluded, retaining only everything derived from the 12 articles finally included according to the designed criteria.

 Reviewer 3 Table 1: The number of studies presented is eleven. Does one of these studies not have a theoretical framework? Each of the studies presents a theoretical framework, through which each of the hypotheses was established for the validation of the theoretical model that, therefore, would be analyzed in the present systematic literature review. Furthermore, based on these theoretical frameworks, an adoption model that includes the main models and variables of the research is proposed in the "theoretical contributions" section.

 Reviewer 3 Minor issues

Figure 7 can benefit from higher resolution as it is hard to read unless the paper is zoomed in.

ICT on page 15 is mentioned without spelling the entire definition before using the acronym. Figure 7 is eliminated, as part of the restructuring of the bibliometric elements of the article. In addition, the explanation of the acronym TIC is added to the page in question.

 Reviewer 3 The methodology lacks many best practices in systematic reviews. The search process shows potential bias in the datasets used for this review. The authors do not present a table summarising the results (a matrix table) that includes all the studies with data extracted from each primary study. This table is a standard practice in systematic reviews that allows the reader to follow the different aspects of each study. Moreover, as in Table 1, the number of studies is less than the final number. A comprehensive table of features extracted allows better follow-up and data summarising. A comprehensive table can be used for producing summary statistics. Another major issue is the analysis of 180 studies in certain parts and then shifting to the primary 12 studies in other parts. The discussion is easy to follow and well-written. On the one hand, the subsection "assessment of the risk of bias" of the research is added. A matrix table is also presented that accounts for the 12 studies included in the systematic literature review. Likewise, the frequency is added to Table 1 (Then Table 2, after adding the summary table) accounting for the number of studies that have validated each theory, this being supported by the summary table. Finally, according to several recommendations, the bibliometric phase is excluded, where 180 articles were analyzed, and only the 12 articles that passed the 3 exclusion phases were analyzed.

We look forward to your comments and hope to hear from you soon.

Thank you very much

_

The authors

---

## [Decision Letter · Decision Letter 1]

25 Sep 2023

Smart home adoption factors: A systematic literature review and research agenda

PONE-D-23-13511R1

Dear Dr. Valencia-Arias,

We’re pleased to inform you that your manuscript has been judged scientifically suitable for publication and will be formally accepted for publication once it meets all outstanding technical requirements.

Kind regards,

Mohammed A. Al-Sharafi

Academic Editor

PLOS ONE

Additional Editor Comments (optional):

Reviewers' comments:

Reviewer's Responses to Questions

**Comments to the Author**

1. If the authors have adequately addressed your comments raised in a previous round of review and you feel that this manuscript is now acceptable for publication, you may indicate that here to bypass the “Comments to the Author” section, enter your conflict of interest statement in the “Confidential to Editor” section, and submit your "Accept" recommendation.

Reviewer #1: All comments have been addressed

Reviewer #3: All comments have been addressed

2. Is the manuscript technically sound, and do the data support the conclusions?

Reviewer #1: Yes

Reviewer #3: Yes

3. Has the statistical analysis been performed appropriately and rigorously? 

Reviewer #1: N/A

Reviewer #3: Yes

4. Have the authors made all data underlying the findings in their manuscript fully available?

Reviewer #1: Yes

Reviewer #3: Yes

5. Is the manuscript presented in an intelligible fashion and written in standard English?

Reviewer #1: Yes

Reviewer #3: Yes

6. Review Comments to the Author

Reviewer #1: Dear Author,

The researchers stated that smart homes represent the complement of various automation technologies that together make up a network of devices facilitating the daily tasks of residents. These technologies are being studied for their application from different sectors, including the projection of their use to improve energy consumption planning and health care management. However, technology adoption depends on social awareness within the scope of cognitive advantages and innovations compared to perceived risk because although there are multiple benefits, potential users express fears related to the loss of autonomy and security. This study carries out a systematic literature review based on PRISMA in order to analyze research trends and literary evolution in the technological adoption of smart homes, considering the main theories and variables applied by the community. In proposing a research agenda in accordance with the identified gaps and the growing and emerging themes of the object of study, it is worth highlighting the growing interest in the subject, both for the present and its development in the future. Until now, adoption factors have been attributed more to the technological acceptance model and the diffusion of innovation theory, adopting components of the Theory of Planned Behavior; therefore, in several cases, the attributes of different theories are merged to adapt to the needs of each researcher, promoting the creation of empirical and extended models. Indeed, the current revised paper shows a proper enhancement on the relevant literature, methodology, and most importantly the discussion and conclusions.

Reviewer #3: The response provided by the authors to my comments is satisfactory. There are still some improvements regarding the potential bias in their search methodology. However, they tried to explain their choices and addressed this potential bias as a limitation. They removed the figures that were unnecessary as recommended. Also, I find a consolidated table of attributes of the studies would have been more beneficial that the current employed tabulation in the study. However, I find following the logic comprehensible. As a final decision, I feel the manuscript in its current form is suitable for publication.

7. PLOS authors have the option to publish the peer review history of their article (what does this mean?). If published, this will include your full peer review and any attached files.

Reviewer #1: No

Reviewer #3: **Yes: **Osama Abdelhay

---

## [Editor Report · Acceptance letter]

12 Oct 2023

PONE-D-23-13511R1 

Smart home adoption factors: A systematic literature review and research agenda 

Dear Dr. Valencia-Arias:

I'm pleased to inform you that your manuscript has been deemed suitable for publication in PLOS ONE. Congratulations! Your manuscript is now with our production department. 

Kind regards, 

on behalf of

Dr. Mohammed A. Al-Sharafi 

Academic Editor

PLOS ONE